# A Heterogeneous Graph Optimization Perspective for Multi-Agent System Workflows

## Abstract

Large Language Model (LLM)-based multi-agent systems (MAS) have shown potential in solving complex tasks across a wide range of domains. However, designing effective MAS workflows remains a significant challenge. Manually crafted workflows are difficult to scale and adapt. Automated workflow optimization techniques usually depend heavily on the planning capability of meta-agent, cannot fully utilize the historical context, and neglect the dynamic interactions between agents and tools. To address these limitations, we propose **He**terogeneous **G**raph-based work**Flow** optimization (HeGFlow), which models agents, tools, and reasoning steps as interconnected graph components, transforming the design of MAS workflow into a heterogeneous graph adjacency matrix optimization problem. To efficiently explore the vast search space, HeGFlow introduces a two-stage matrix training process guided by a subgraph sampling strategy. Extensive experiments across six complex domains show that HeGFlow enables smaller LLMs to match or even surpass the performance of much larger models. Furthermore, HeGFlow consistently outperforms existing manually and automated workflow approaches on four widely-used benchmarks, establishing a new paradigm for scalable and effective MAS workflow generation.

## 1 Introduction

Large Language Model (LLM)-based multi-agent systems (MAS), which assign distinct roles and capabilities to different agents and enable their interactions, have been applied to a wide range of domains, such as scientific research (Gottweis et al., 2025; Ghareeb et al., 2025), software development (Hong et al., 2023), and long-form writing (Wang et al., 2024b; Zheng et al., 2023).

A key research challenge for MAS is to design and optimize the workflow, where a workflow is a structured sequence of agent invocations (Zhang et al., 2025b). Early approaches, such as AutoGen (Wu et al., 2023) and CAMEL (Li et al., 2023), depend on manual design and careful preconfiguration for specific tasks. This labor-intensive paradigm inherently limits their adaptability across diverse domains.

Therefore, research has increasingly shifted towards automating MAS workflow design. Initial efforts include prompt-level optimization (Khattab et al., 2023; Yang et al., 2023), inter-agent communication topology optimization (Zhuge et al., 2024; Zhang et al., 2024a; Wang et al., 2025), and agent profiling improvement (Saad-Falcon et al., 2024; Yuan et al., 2024; Chen et al., 2023), etc. More recent approaches formulate MAS workflow optimization as a search problem within the agent design space. For example, AgentSquare (Shang et al., 2024) modularizes agent design, where the module-programming LLM generates new modules based on historical experience and current task. ADAS (Hu et al., 2025) and AFlow (Zhang et al., 2025b) use a meta-agent to generate and optimize workflows. Despite these advances, current methods still face key challenges: (1) They heavily rely on the capability of a meta-agent to generate new agents or workflows, which increases the risk of getting stuck in local optima, as well as leads to significant resource overhead; (2) Lacking an effective memory mechanism, these approaches struggle to fully leverage historical experience; (3) They often overlook tools as a key component, resulting in inadequate optimization of agent-tool interactions.

To address these limitations, we propose HeGFlow, a novel framework for MAS workflow optimization based on heterogeneous graphs. Specifically, we model MAS as a heterogeneous graph,

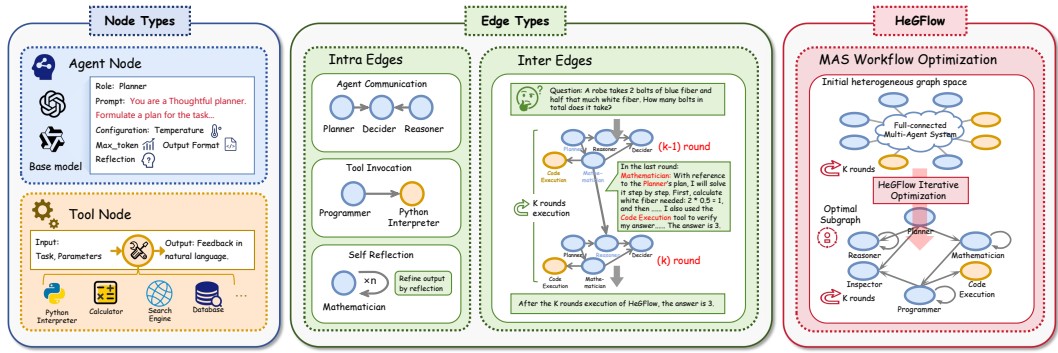

Figure 1: Nodes, edges and the optimization objective of HeGFlow.

where agents and tools are nodes, and edges represent the interactions between agents, tool invocations, and self-reflections. In this way, we transform the MAS workflow generation into the graph adjacency matrix optimization problem. With this formulation, feedback during the optimization process is applied directly to the graph structure as a reward signal and is then reflected in the matrix update. This mechanism not only leverages the entire historical experience of the training process, but also facilitates fine-grained optimization, which is often challenging to achieve with existing natural language feedback. However, to enable efficient exploration of the optimal graph matrix, it faces the issues of efficiently navigating the vast heterogeneous graph search space and slow convergence. Therefore, we introduce a novel two-stage graph matrix training approach, which consists of coarse-grained node selection followed by fine-grained edge optimization. To strike a robust balance between exploration and exploitation in the graph search space, we design a subgraph sampling policy, which allows for effective utilization of global information and thereby preventing premature convergence to local optima. Crucially, our MAS workflow represented by a sampled subgraph is a decentralized structure, and agent nodes can be instantiated using small models. This design choice significantly reduces computational costs and effectively decouples the system's reliance on large models. Our main contributions are as follows:

- We introduce a novel framework that models MAS as a heterogeneous graph, transforming the problem of automating MAS workflow into adjacency matrix optimization. This can facilitate a fine-grained optimization and fully leverage historical experience through adjacency matrix updates.

- We develop a two-stage graph matrix training, integrating coarse-grained node selection and fine-grained edge optimization to enhance efficiency. This improves efficiency in a large search space and strikes a balance between exploitation and exploration.

- Extensive empirical validations show that our approach can perform comparable to or even exceed larger models when only using smaller models, and outperforms state-of-the-art automated workflow optimization methods.

## 2 METHODOLOGY

Our approach works by modeling a multi-agent system as a heterogeneous graph, thereby converting the MAS workflow generation into an adjacency matrix optimization problem. However, directly optimizing the full matrix is challenging due to the vast search space, which may lead to slow convergence and the potential local optima. To overcome this, we propose a two-stage training strategy to improve the efficiency of node selection and edge optimization.

### 2.1 HETEROGENEOUS GRAPH MODELING

A multi-agent system is modeled as a heterogeneous graph $\mathcal{G} = \{\mathcal{V}, \mathcal{E}\}$, where $\mathcal{V}$ is the set of nodes, and $\mathcal{E}$ represents the set of edges between these nodes. As shown in Fig. 1, nodes include

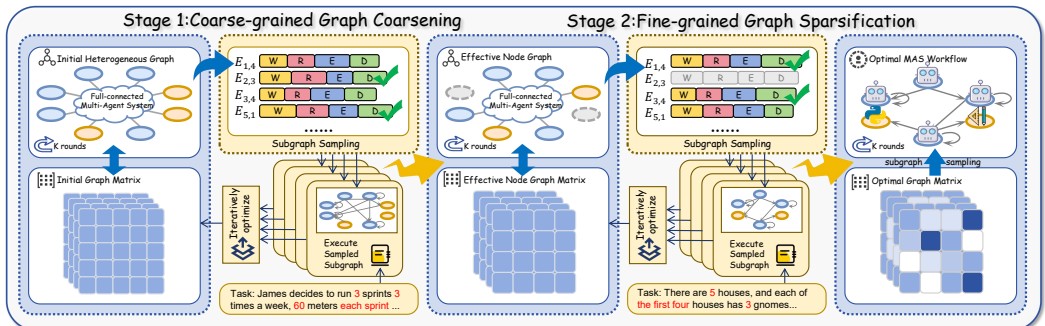

Figure 2: Two-stage heterogeneous graph matrix training. Stage 1: Iteratively sample and evaluate subgraphs for coarse-grained graph coarsening, removing ineffective nodes (grayed nodes). Stage 2: Based on the graph matrix obtained from Stage 1, optimize the graph matrix to recognize the effective interaction edges in a fine-grained way (depth of color in the matrix represents edge weight). After the two stages, an optimized heterogeneous graph matrix is obtained, from which the optimal subgraph can be sampled with the highest probability.

agents and tools, i.e., $\mathcal{V} = \mathcal{V}_{agent} \cup \mathcal{V}_{tool}$. Each agent node is characterized by $\Phi = (M, P, R)$, where $M$ denotes the underlying base model; $P$ is the prompt that includes the definition of the role, the description of the task, the output of the related agents, and the output format; $R$ indicates that the agent reflects on its output. Meanwhile, each tool node is characterized by the type of tool. Tool nodes serve as a crucial component for extending agent capabilities by encapsulating external APIs or specialized utilities (e.g., code interpreters, web search engines, calculator and database interfaces).

An edge formalizes the various interactions among the nodes, including information propagation, collaborative dependencies, and result feedback. Each edge is parameterized by a learnable weight $W^{(e)}$ that indicates the significance or intensity of the modeled interaction. In order to harness the agent's self-correction ability, our system performs $K$ iterative rounds per query, where the outputs from round $(k-1)$ are forwarded to the nodes in round $k$, which allows these nodes to refine their behavior in light of the previous round's results.

Based on this multi-round setting, edges are classified as intra-round edge $\mathcal{E}_{intra}$ and inter-round edge $\mathcal{E}_{inter}$. Consequently, we define two types of adjacency matrices: intra- round edge adjacency matrix $\mathbf{A}_{intra}$, and inter-round edge adjacency matrix $\mathbf{A}_{inter}$. $\mathbf{A}_{intra}^{(k)}(i,j) \in [0,1]$ indicates the connection strength from nodes $v_i$ to $v_j$ in round $k$. In particular, $\mathbf{A}_{intra}^{(k)}(i,i) \neq 0$ means that there is a self-loop for node $v_i$ in round $k$, representing the agent's reflective information flow. $\mathbf{A}_{inter}^{(k)}(i,j) \in [0,1]$ means the intensity that the output from node $v_i$ in the $(k-1)$-th round is passed to node $v_j$ in the $k$-th round. Further, we define the intra/inter neighbors for each node as follows:

$$\mathcal{N}_{intra}^{(k)}(v_i) = \left\{ v_j \mid \mathbf{A}_{intra}^{(k)}(j,i) \neq 0 \right\}, \mathcal{N}_{inter}^{(k)}(v_i) = \left\{ v_j \mid \mathbf{A}_{inter}^{(k)}(j,i) \neq 0 \right\}. \quad (1)$$

Therefore, for each agent node $v_i$ in round $k$, its output $O_i^{(k)}$ will be collectively determined by task $\mathcal{T}$, its own configuration $\Phi_i$, and output from intra/inter-round neighbors, as formalized below:

$$O_i^{(k)} \sim \mathcal{F}\left( \mathcal{T}, \Phi_i, \cup_{v_j \in \mathcal{N}_{intra}^{(k)}(v_i)} O_j, \cup_{v_j \in \mathcal{N}_{inter}^{(k)}(v_i)} O_j \right). \quad (2)$$

By modeling MAS as a heterogeneous graph $\mathcal{G}$ parameterized by a set of adjacency matrices $\mathbf{A} = \mathbf{A}_{intra} \cup \mathbf{A}_{inter}$, given a task $\mathcal{T}$, we use $\mathcal{O}(\mathbf{A}, \mathcal{T})$ to denote the utility function, which measures system performance and can be obtained by sampling and executing a workflow from $\mathbf{A}$. The objective is to find the optimal parameterized $\mathbf{A}^*$, which can achieve the highest probability of sampling an optimal workflow. However, this endeavor poses the following key challenges: achieving fast convergence while avoiding local optima in a vast search space; ensuring efficient sampling with a well-balanced trade-off between exploration and exploitation.

## 2.2 OPTIMAL SUBGRAPH GENERATION

To address these challenges, we propose two key strategies: a subgraph sampling policy and a two-stage graph matrix training approach.

### 2.2.1 SUBGRAPH SAMPLING POLICY

Subgraph sampling plays a critical role by continuously exploring the graph search space, where performance on the task provides feedback to iteratively optimize the graph adjacency matrix $\mathbf{A}$. The subgraph sampling strategy should ideally meet two key objectives: (1) Ensuring that each sampled subgraph is a valid workflow; and (2) Balancing exploration and exploitation to effectively navigate the graph search space.

To ensure that the sampled subgraphs are valid workflows, we consider two categories of intra-edges. For diagonal edges that represent reflections, we directly compute the sampling probability based on the edge weight to determine whether an agent node should undergo reflection. For non-diagonal edges, we follow the usual setup to ensure that the graph structure forms a directed acyclic graph (DAG) (Zhuge et al., 2024). This guarantees a valid execution order in the sampled subgraph. The detailed algorithm is presented in Appendix D.

Next, we consider how to compute the selection probability for each edge. Traditional graph sampling algorithms typically determine sampling probabilities based solely on the learned edge weights. While straightforward, this approach often leads to premature convergence or is trapped in local optima, as it predominantly favors exploitation of currently high-weighted edges without sufficient exploration of potentially beneficial, yet less frequently visited, graph structures. The Upper Confidence Bound (UCB) algorithm is usually used to address the exploration-exploitation trade-off in dynamic decision-making (Auer et al., 2002). Inspired by this, we introduce a graph sampling policy to dynamically balance the trade-off between exploration (discovering new, potentially optimal edges) and exploitation (leveraging currently known high-performing edges), thereby enabling a more robust and global optimization of the graph. With our sampling algorithm, the weight $W_s^{(e)}$ of an edge $e$ being selected is:

$$
W_s^{(e)} = \underbrace{\alpha W^{(e)} + \beta \frac{N_{cor}^{(e)}}{N_{sel}^{(e)}}}_{\text{exploitations}} + \underbrace{\gamma \sqrt{\frac{2\log\left(N_{sel}^{all}+1\right)}{N_{sel}^{(e)}}} + \frac{\delta}{N_{sel}^{(e)}+1}}_{\text{explorations}}, \tag{3}
$$

where $\alpha$, $\beta$, $\gamma$, and $\delta$ are coefficients. The first term $\alpha W^{(e)}$, represents the weight of edge $e$, which is optimized during training and effectively captures the utility of the connection. The second term, $\beta \frac{N_{cor}^{(e)}}{N_{sel}^{(e)}}$, represents the average reward for edge $e$, where $N_{cor}^{(e)}$ is the number of correct answers for this edge, and $N_{sel}^{(e)}$ is the times that the edge is sampled. This term ensures that edges with higher historical performance are prioritized. Therefore, the first two terms serve as exploitations, ensuring that both the reward and effectiveness of each edge are properly considered. The third term, $\gamma \sqrt{\frac{2\log\left(N_{sel}^{all}+1\right)}{N_{sel}^{(e)}}}$, serves as the exploration component, encouraging the selection of edges with fewer trials, where $N_{sel}^{all}$ represents the total number of edges selected so far. The fourth term, $\frac{\delta}{N_{sel}^{(e)}+1}$, is a diversity bonus, which tries to avoid over-utilization of some edges. The last two terms constitute the exploitation term, ensuring that under-explored edges are adequately explored, thus mitigating the possibility of settling in local optima.

### 2.2.2 TWO-STAGE HETEROGENEOUS GRAPH MATRIX TRAINING

Directly optimizing the graph matrix will result in a large search space, leading to slow convergence and significant computation overhead due to redundant explorations. To address this issue, we propose a two-stage subgraph optimization approach. The first stage, a coarse-grained graph coarsening, prioritizes quick node selection, deferring fine-grained optimization of inter-node interaction edges. Subsequently, the second stage, a fine-grained graph sparsification, focuses on optimizing

---

**Algorithm 1** Two-stage graph matrix training

---

**Require**: Initial heterogeneous graph $\mathcal{G}$, initial graph matrix $\mathbf{A} = \mathbf{A}_{intra} \cup \mathbf{A}_{inter}$, set of tasks $\mathcal{T}$, number of sampled subgraphs each time $N$, number of iterations for each stage $T_1$ and $T_2$
**Ensure**: Optimal graph matrix $\mathbf{A}^*$

1: **Stage 1**: Coarse-grained node selection
2: Initialize the heterogeneous graph matrix $\mathbf{A}^{(0)}$
3: **for** $t = 1$ to $T_1$ **do**
4:     Sample $N$ subgraphs $\{\mathcal{G}_i^{(t)}\}_{i=1}^N$ from $\mathbf{A}^{(t-1)}$
5:     Execute task $\mathcal{T}^{(t)}$ based on subgraph $\{\mathcal{G}_i^{(t)}\}_{i=1}^N$.
6:     $\mathbf{A}^{(t)} \leftarrow \mathbf{A}^{(t-1)} + \eta_1 \cdot (\nabla_{\mathbf{A}} \mathcal{O}(\mathcal{G}_i^{(t)}) - \sigma_1 \cdot \sum_k \nabla_{\mathbf{A}} \left\| \mathbf{A}^{(\mathbf{k})} \right\|_* - \lambda \cdot \nabla_{\mathbf{A}} \mathcal{L}_{sparse}^{(t)})$
7: **end for**
8: $\tilde{\mathcal{G}} \leftarrow \mathcal{G}, \tilde{\mathbf{A}} \leftarrow \mathbf{A}$
    // Determine the valid nodes from the graph.
9: **Stage 2**: Fine-grained edge optimization
10: Initialize the heterogeneous graph matrix $\tilde{\mathbf{A}}^{(0)}$
11: **for** $t = 1$ to $T_2$ **do**
12:     Sample $N$ subgraphs $\{\tilde{\mathcal{G}}_i^{(t)}\}_{i=1}^N$ from $\tilde{\mathbf{A}}^{(t-1)}$
13:     Execute task $\mathcal{T}^{(t)}$ based on subgraph $\{\tilde{\mathcal{G}}_i^{(t)}\}_{i=1}^N$.
14:     $\tilde{\mathbf{A}}^{(t)} \leftarrow \tilde{\mathbf{A}}^{(t-1)} + \eta_2 \cdot (\nabla_{\mathbf{A}} \mathcal{O}(\mathcal{G}_i^{(t)}) - \sigma_2 \cdot \sum_k \nabla_{\mathbf{A}} \left\| \tilde{\mathbf{A}}^{(k)} \right\|_*$
15: **end for**
16: Obtain the optimal graph matrix $\mathbf{A}^*$ after the two stages.
17: **return** $\mathbf{A}^*$

---

these edges within the selected set of nodes from the first stage. After these two stages, we obtain the heterogeneous graph matrix that maximizes the probability of sampling the optimal subgraph.

To maximize system performance, due to the non-differentiability of the utility function $\mathcal{O}(\cdot)$, we use the probability-weighted average performance of a limited number of samples to approximate the overall performance expectation, i.e.,

$$\nabla_{\mathbf{A}} E_{\mathcal{G}' \sim S(\mathbf{A})} \left[ \mathcal{O}\left( \mathcal{G}', \mathcal{T} \right) \right] \approx \frac{1}{N} \sum_{n=1}^N \mathcal{O}\left( \mathcal{G}_n, \mathcal{T} \right) \nabla_{\mathbf{A}} \log \left( P_{\mathbf{A}}^{\mathcal{G}_n} \right), \tag{4}$$

where $P_{\mathbf{A}}^{\mathcal{G}_n}$ represents the probability of sampling subgraph $\mathcal{G}_n$ from $\mathbf{A}$ by the subgraph sampling policy.

Note that in the first stage, our objective is to rapidly identify the high-performance nodes. Therefore, in addition to maximizing performance, we also incorporate node sparsity into the reinforcement learning optimization objective. This is facilitated by a penalty term $\mathcal{L}_{sparse}$ that targets the minimization of connection strengths of the neighboring edges of inefficient nodes,

$$\mathcal{L}_{sparse} = \sum_i w_i \cdot \left( -\log \left( 1 - \mu_{v_i} \right) \right), \tag{5}$$

where $\mu_{v_i}$ is the average weights of all intra-edges connected to node $v_i$. $w_i$ is a scaling factor, where for an inefficient node, we assign a larger $w_i$ to amply its loss, and for an efficient node, we assign a smaller $w_i$, allowing them to be preserved during training. In this way, we enable a quick reduction of edge weights of inefficient nodes and promote the selection of high-performing nodes. To further accelerate training, we also consider low-rank sparsity, which ensures that learned connectivity remains sparse and robust, and eliminate redundant and inefficient interactions (Entezari et al., 2020). Following the approach of Zhang et al. (2024a), we use the nuclear norm $\left\| \mathbf{A}^{(\mathbf{k})} \right\|_*$. Therefore, the overall objective for the first stage can be expressed as:

$$\max_A \mathbb{E}_{\mathcal{G}' \sim S(\mathbf{A})} \mathcal{O}(\mathcal{G}', \mathcal{T}) - \sigma \sum_k \left\| \mathbf{A}^{(\mathbf{k})} \right\|_* - \lambda \mathcal{L}_{sparse}(\mathbf{A}), \tag{6}$$

where $\sigma$ and $\lambda$ are coefficients.

Through such coarse-grained training, the average connection strength of inefficient nodes will become extremely low. We remove these nodes and obtain the intermediate adjacency matrix $\tilde{\mathbf{A}}$ that consists of all the agent and tool nodes required for the task.

For the second stage, fine-grained graph sparsification, the objective is to optimize the interaction edges in the graph, increasing the weights of effective interaction edges while reducing that of inefficient or redundant edges. Hence, this stage focuses more on the nuanced adjustment of the heterogeneous graph matrix from the first stage. Therefore, different from the first stage, the objective for the second stage does not use the penalty $\mathcal{L}_{sparse}$:

$$\max_{\tilde{A}} \mathbb{E}_{\mathcal{G}' \sim S(\tilde{\mathbf{A}})} \mathcal{O}(\mathcal{G}', \mathcal{T}) - \sigma \sum_k \left\| \tilde{\mathbf{A}}^{(\mathbf{k})} \right\|_* . \tag{7}$$

The training process is summarized in Algorithm 1. After the two-stage matrix training, we obtain an optimized heterogeneous graph adjacency matrix, which promotes efficient interactions by assigning high weights to connections between effective nodes, thereby maximizing the probability of sampling an optimal subgraph. For inference, given task $\mathcal{T}$, we sample a subgraph as a workflow using the subgraph sampling policy.

Table 1: Performance of HeGFlow and baselines across six domains on the MMLU-Pro benchmark. The best results are in **bold**, and the second-best results are underlined.

| Model | Math | Physics | Eng. | History | Law | Psychology | Avg. |
|---|---|---|---|---|---|---|---|
| **Open-source Models** | | | | | | | |
| Llama-3-8B-Instruct | 36.1 | 34.4 | 31.3 | 42.3 | 26.5 | 59.4 | 38.3 |
| DeepSeek-V2-Chat | 53.7 | 54.0 | 31.9 | 45.3 | 40.6 | 66.2 | 48.6 |
| Phi-3-medium-4k-instruct | 52.2 | 49.4 | 37.9 | 57.2 | 38.3 | 73.4 | 51.4 |
| Llama-3-70B-Instruct | 54.0 | 49.6 | 43.6 | 56.9 | 39.9 | 70.2 | 52.4 |
| Qwen3-8B | 75.1 | 62.1 | 50.8 | 54.9 | 35.1 | 63.5 | 56.9 |
| DeepSeek-V3-0324 | 92.3 | 76.9 | 57.3 | 66.4 | 52.1 | 67.3 | 68.7 |
| **Closed-source Models** | | | | | | | |
| GPT-4-Turbo | 62.8 | 61.0 | 35.9 | 67.7 | 51.2 | 78.3 | 59.5 |
| Claude-3-Opus | 69.6 | 69.7 | 48.4 | 61.4 | **53.5** | 76.3 | 63.2 |
| Gemini-1.5-Pro | 72.8 | 70.4 | 48.7 | 65.6 | 50.8 | 77.2 | 64.3 |
| GPT-4o | 76.1 | 74.7 | 55.0 | **70.1** | 51.0 | **79.2** | 67.7 |
| **Multi-Agent Frameworks** | | | | | | | |
| AutoGen | 80.2 | 59.2 | 49.5 | 50.6 | 28.1 | 68.1 | 56.0 |
| CAMEL | 87.5 | 71.0 | 55.7 | 53.2 | 38.9 | 66.9 | 62.2 |
| HeGFlow (Ours) | **93.0** | **77.6** | **57.5** | 61.5 | 52.3 | 74.1 | **69.3** |

## 3 EXPERIMENTS

In this section, we aim to assess HeGFlow's adaptability and overall performance across diverse multi-task scenarios. Our investigations attempt to address the following research questions. **RQ1**: Can HeGFlow demonstrate robust and superior performance in language comprehension and reasoning across diverse domains? **RQ2**: How does HeGFlow's performance compare with existing manually and automated workflow optimization method? **RQ3**: Can HeGFlow achieve consistent performance improvements when varying scales or architectures of large language models are applied? **RQ4**: Can HeGFlow's ability in using tools lead to performance gains compared to the conventional prompt-based or ReAct paradigms (Yao et al., 2023)? **RQ5**: What is HeGFlow's token consumption as compared with other approaches?

### 3.1 MULTI-DOMAIN TASKS PERFORMANCE (RQ1)

#### 3.1.1 SETTINGS

To address RQ1, we choose the MMLU-Pro dataset (Wang et al., 2024a), a challenging benchmark comprising language comprehension and reasoning questions across 14 different domains,

with each question having 10 options. This dataset is significantly more demanding than its predecessor, MMLU (Hendrycks et al., 2020), as it rigorously evaluates both the system's knowledge acquisition and sophisticated reasoning capabilities. For our evaluation, we focus on six representative domains: Mathematics, Physics, Engineering, History, Law, and Psychology. We compare HeGFlow with three types of baseline: **(1) Open-source models:** including Llama-3-8B-Instruct and Llama-3-70B-Instruct (Grattafiori et al., 2024), DeepSeek-V2-Chat (Liu et al., 2024a), Phi-3-medium-4k-instruct (Abdin et al., 2024), Qwen3-8B (Yang et al., 2025) and DeepSeek-V3-0324 (DeepSeek-AI, 2024); **(2) Closed-source models:** including GPT-4-Turbo (Achiam et al., 2023), Claude-3-Opus (Anthropic, 2025), Gemini-1.5-Pro (Google, 2024) and GPT-4o (OpenAI, 2024); **(3) Multi-agent framework:** including AutoGen (Wu et al., 2023) and CAMEL (Li et al., 2023), where both these two frameworks and HeGFlow use Qwen3-8B as base model.

Table 2: Comparisons with manually designed and automated workflow optimization methods across four distinct benchmarks. For all approaches, we set GPT-4o-mini as the base model. The best results are highlighted in **bold**.

| Method | Metrics | | | | |
|---|---|---|---|---|---|
| | DROP F1 Score | MGSM Accuracy(%) | GSM8K Accuracy(%) | MATH Accuracy(%) | Avg. |
| **Manually Designed Methods** | | | | | |
| Vanilla (GPT-4o-mini) | 68.3 | 80.9 | 92.7 | 46.3 | 72.1 |
| Chain-of-Thought | 78.5 | 83.4 | 92.4 | 46.4 | 75.2 |
| CoT-SC | 78.8 | 84.5 | 92.7 | 47.9 | 76.0 |
| Self Refine | 70.2 | 81.6 | 89.6 | 46.1 | 71.9 |
| LLM Debate | 78.9 | 86.7 | 89.5 | 48.5 | 75.9 |
| MultiPersona | 74.4 | 86.1 | 92.8 | 45.4 | 74.7 |
| **Automated Workflow Optimization Methods** | | | | | |
| GPTSwarm (Zhuge et al., 2024) | 78.1 | 86.1 | 93.3 | 47.9 | 76.4 |
| ADAS (Hu et al., 2025) | 76.6 | 81.2 | 90.8 | 43.2 | 73.0 |
| AFlow (Zhang et al., 2025b) | 80.6 | 89.7 | 93.5 | 51.3 | 78.8 |
| HeGFlow (Ours) | **81.3** | **91.2** | **94.2** | **52.1** | **79.7** |

### 3.1.2 RESULTS

As shown in Table 1, the results demonstrate the remarkable adaptability of HeGFlow across diverse domains. Our approach, while using small base models, matches or surpasses the performance of significantly larger models. Importantly, when using the same base model (e.g., Qwen3-8B), its performance consistently exceeds that of other multi-agent frameworks (AutoGen and CAMEL). These findings validate that our heterogeneous graph based framework provides a robust mechanism for effectively organizing and leveraging multi-agent capabilities to tackle a wide array of complex tasks. HeGFlow is capable of identifying complex agent collaborations that have been overlooked by manual designs, thereby better unleashing the reasoning potential of agents. Regarding the suboptimal performance of our system in the History, Psychology, and Law domains, we posit that for such knowledge-intensive tasks, the integration of simple retrieval tools with multi-agent collaborative analysis may not fully mitigate the impact of model parameter scale and training data volume.

## 3.2 COMPARISONS WITH EXISTING WORKFLOW GENERATION METHODS (RQ2)

### 3.2.1 SETTINGS

To address RQ2, We evaluate HeGFlow on four public benchmarks covering three domains: **(1) Reading comprehension**, DROP (Dua et al., 2019); **(2) Math reasoning**, GSM8K (Cobbe et al., 2021) and MATH (Hendrycks et al., 2021); **(3) Multi-language comprehension**, MGSM (Shi et al., 2022). For MATH, we follow Hong et al. (2024) in selecting 617 problems from four typical problem types (Combinatorics & Probability, Number Theory, Pre-algebra, Pre-calculus) at difficulty level 5. We consider two types of baselines: **(1) Manually designed methods**: Including Chain-of-Thought (Wei et al., 2022), Self-Consistency with Chain-of-Thought (CoT-SC) (Wang et al., 2022), Self-refine (Madaan et al., 2023), LLM-Debate (Du et al., 2024), and MultiPersona Debate (Wang et al., 2023). **(2) Automated workflow optimization methods:** Including GPTSwarm (Zhuge et al.,

2024),ADAS (Hu et al., 2025) and AFlow (Zhang et al., 2025b). Since the translation tool is not used for other frameworks on the MGSM dataset, to ensure a fair comparison, we do not employ it as well.

### 3.2.2 RESULTS

The results in Table 2 show that HeGFlow achieves the best average performance of 79.7, and surpasses all manual design methods by an average of 3.7~7.8% and outperforms the state-of-the-art automatic workflow optimization methods by an average of 0.9~6.7%, demonstrating its capability and adaptability. Specifically, HeGFlow performs better than similar work on MATH, which only retains the highest level of difficulty, indicating that the workflow generated by HeGFlow can more effectively enhance the reasoning and decision-making capabilities of MAS.

Table 3: Performance on GSM8K with different base models

| Base Model | Methods | | | | |
|---|---|---|---|---|---|
| | Vanilla | CoT | CoT-SC | AutoGen | HeGFlow |
| Llama3-8B | 55.3 | 61.2 | 65.6 | 63.8 | **76.1** |
| Qwen3-8B | 89.8 | 91.1 | 91.0 | 93.2 | **95.5** |
| GPT-4o-mini | 92.7 | 92.4 | 92.7 | 92.9 | **94.2** |

### 3.3 COMPARISONS OF DIFFERENT BASE MODELS (RQ3)

### 3.3.1 SETTINGS

To answer RQ3, we evaluate the adaptability of our method across different base models. The experiments are conducted on GSM8K, and the baselines include vanilla that refers to direct model inference without any specialized prompt, Chain-of-Thought (CoT), CoT with Self-Consistency (CoT+SC) and AutoGen. The evaluation is performed using three different base models: Llama3-8B, Qwen3-8B, and GPT-4o-mini.

### 3.3.2 RESULTS

The results presented in Table 3 demonstrate that our framework achieves substantial performance gains with different underlying models, irrespective of their inherent capacity. Especially for Llama-3-8B, compared with the vanilla base model, HeGFlow achieves a 20.8% improvement in accuracy. Crucially, HeGFlow consistently surpasses all baseline methods, highlighting its robust generalizability and model-agnostic applicability.

### 3.4 THE ABILITY OF USING TOOLS (RQ4)

### 3.4.1 SETTINGS

To answer RQ4, we compare against ReAct (Yao et al., 2023), a prevailing agentic framework in which agents continuously perceive the environment and execute the corresponding actions. We conduct this evaluation on the MGSM and MATH datasets. MGSM is a mathematical dataset consisting of 10 different languages, and models' suboptimal performance on it primarily stems from language barriers that hinder accurate problem analysis. For the MGSM dataset, we specifically incorporate translator and calculator tools, which are also integrated as tool nodes within our framework. Meanwhile, for the MATH dataset, we introduce calculator and Python code execution tools. For this experiment, we employ two distinct base models: Llama3-8B, characterized by weaker reasoning capability, and Qwen3-8B, which exhibits stronger reasoning. Larger models are excluded, as their extensive multilingual pre-training might diminish the impact of explicit tool utilization.

### 3.4.2 RESULTS

The results are shown in Fig. 3. On the MATH benchmark, compared with ReAct, our approach has respectively brought about a 2.5% and 4.5% increase in accuracy on the two base models. On the MGSM dataset, our method outperforms ReAct by 20.8% and 7.4% in accuracy on the two base models. Apparently, the translation tool is beneficial in solving problems in MGSM. For the Llama3-8B model, the inefficiency of the ReAct paradigm stems from its limited context and reasoning capabilities. In

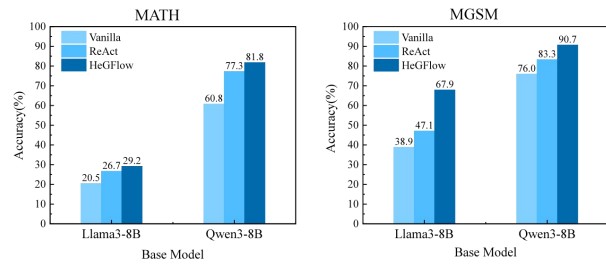

Figure 3: Performance of tool use on MATH and MGSM

contrast, Qwen3-8B often tends to be overconfident in its language capabilities, leading it to underutilize external tools. HeGFlow addresses this issue by orchestrating sequential node invocation, effectively mitigating the limitations of relying on a single model's decision-making.

### 3.5 COST ANALYSIS (RQ5)

#### 3.5.1 SETTINGS

To demonstrate that HeGFlow is cost-efficient, we consider token consumption in both training and inference, and compare HeGFlow with other MAS workflow approaches, including LLM-Debate, DyLAN (Liu et al., 2024b), GPTSwarm (Zhuge et al., 2024), and AFlow.

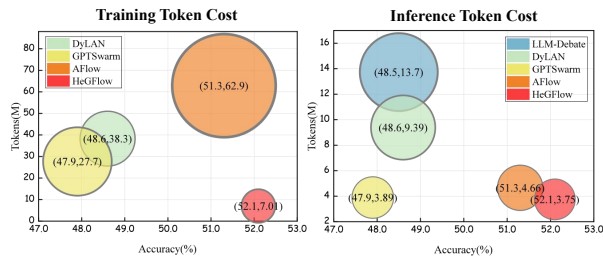

Figure 4: Visualization of performance and token consumption on MATH benchmark.

#### 3.5.2 RESULTS

As shown in Fig. 4, token consumption includes the training token cost and the inference token cost, measured in millions (M). The diameter of each point is proportional to its y-axis value. We can see that, among these methods for optimizing MAS workflow, HeGFlow has the least total token consumption while achieving the best performance. Especially in the training phase, our proposed two-stage subgraph optimization method can converge quickly, significantly reducing token consumption. GPTSwarm and AFlow have similar token consumption during inference; however, the inefficiency of search space exploration leads to a significant token consumption in the training phase.

## 4 CONCLUSION

This paper presents HeGFlow, a novel heterogeneous graph framework for automated multi-agent system (MAS) workflow optimization. By modeling agents, tools, and reflection mechanisms as distinct node types and edge interactions, we reformulate the design of the MAS workflow as an optimization problem over a heterogeneous graph adjacency matrix. To efficiently explore the large search space, HeGFlow employs a two-stage matrix training process guided by a subgraph sampling strategy. Experiments on five domain-specific benchmarks demonstrate that our framework outperforms both manual and state-of-the-art automated methods. Remarkably, it enables smaller models to match or exceed the performance of much larger ones, offering a scalable and resource-efficient solution for complex tasks.

## 5 ETHICS STATEMENT

We have carefully read and fully adhere to the ICLR Code of Ethics.

## 6 REPRODUCIBILITY STATEMENT

We upload anonymous source code and configuration files as part of the supplementary material to facilitate reproduction of our experiments, and our code is also available at `https://anonymous.4open.science/r/Heterogeneous-graph_MAS-5B85/`. The training procedures are described in Section 2, with additional implementation details provided in Appendix C and F.

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

## A    ACKNOWLEDGMENT OF LARGE LANGUAGE MODEL ASSISTANCE

In the preparation of this manuscript, Large Language Models (LLMs) are used solely for text refinement, including improving clarity, conciseness, and grammar. The use of LLMs does not involve the generation of scientific ideas, interpretations, or data analysis. The authors bear full responsibility for the content, findings, and scientific validity of the work.

## B    RELATED WORK

### B.1    MAS WORKFLOW

Multi-agent system (MAS) workflows represent the predefined execution sequences of agents, which are usually generated based on historical experience and iterative optimization (Zhang et al., 2025b). Early approaches involved manually constructing general workflows (Madaan et al., 2023; Du et al., 2024) and domain-specific workflows such as code generation (Ridnik et al., 2024), data analysis (Xie et al., 2024), and scientific research (Gottweis et al., 2025; Parmar et al., 2025). Although many effective workflows have been developed, it is still challenging to cover the full spectrum of diverse tasks across domains, underscoring the need for automated MAS workflow generation and optimization.

### B.2    AUTOMATING AGENT SYSTEM

Optimizing LLM-based agent systems has been a critical area of research. Representative work include **(1) Prompt Optimization**, such as DSPy (Khattab et al., 2023) and ORPO (Yang et al., 2023); **(2)Agent Profiling**, such as AutoAgents (Chen et al., 2023) and EvoAgent (Yuan et al., 2024); **(3) Inter-agent Communication**, such as GPTSwarm (Zhuge et al., 2024), G-Designer (Zhang et al., 2024b) and AgentPrune (Zhang et al., 2024a). While these methods have demonstrated performance improvements, they often overlook optimization of the entire agentic system. More recent efforts have aimed to optimize complete agent workflows. For instance, AgentSquare (Shang et al., 2024) modularized single-agent workflow, where the workflow is optimized by evaluating different combinations of modules. AFlow (Zhang et al., 2025b) applies Monte Carlo Tree Search over search space, while ADAS (Hu et al., 2025) explores candidates through heuristics-guided expansion, where both of them use a meta-agent to generate and optimize a new agent or workflow in the form of code. However, their reliance on the capabilities of a meta-agent can be a significant bottleneck, and the use of natural language-based historical experience makes them prone to getting stuck in local optima. Furthermore, these approaches frequently overlook critical agentic components, such as reflection and tool integration, which fundamentally constrain the design space and scalability of the resulting systems.

## C    EXPERIMENT DETAILS

In this section, we present details of the experiment, including datasets and baseline setups.

### C.1    DATASETS STATISTICS

We conducted experiments using a total of five datasets, including: MMLU-Pro (Wang et al., 2024a), DROP (Dua et al., 2019), MGSM (Shi et al., 2022), GSM8K (Cobbe et al., 2021) and MATH (Hendrycks et al., 2021). In particular, we utilized six subsets of the MMLU-Pro dataset separately, including Math, Physics, Engineering, History, Law and Psychology. The dataset statistics are presented in Tab. 4.

### C.2    BASELINE SETUP

In this section, we present the detailed settings of the baseline used in the experiments.

Table 4: Datasets Statistics

| Domain | Dataset | #Train | #Test | Metric |
|---|---|---|---|---|
| Language Comprehension and Reasoning | Math | 1013 | 338 | Accuracy |
| | Physics | 974 | 325 | Accuracy |
| | Engineering | 726 | 243 | Accuracy |
| | History | 285 | 96 | Accuracy |
| | Law | 825 | 276 | Accuracy |
| | Psychology | 598 | 200 | Accuracy |
| Reading Comprehension | DROP | 77400 | 9535 | F1 Score |
| Mathematical Reasoning | GSM8K | 7473 | 1319 | Accuracy |
| | MATH | 486 | 119 | Accuracy |
| Multi-language Tasks | MGSM | 1968 | 492 | Accuracy |

### C.2.1 MULTI-DOMAIN TASKS TESTING (RQ1)

In order to improve the performance of open-source models and closed-source models, we follow Wang et al. (2024a) and use 5-shots CoT prompt methods except Gemini-1.5-Pro, which uses 0-shot. Specifically, we add "Let's think step by step" in the prompt to guide the model's thinking and reasoning. Additionally, we provide five examples of correct question-answers for the model to refer to. For the multi-agent frameworks AutoGen (Wu et al., 2023) and CAMEL (Li et al., 2023), we adopt common MAS workflow designs to create customized multi-agent workflows for each task. These workflows typically include key roles such as Planner, Reasoner, Domain Expert, Critic, and Summarizer. It exhibits a hierarchical collaborative structure (Tran et al., 2025) to fully leverage the collective wisdom of multiple agents. We set Qwen3-8B (Yang et al., 2025) as the base model for these two frameworks and our HeGFlow.

### C.2.2 COMPARISONS WITH OTHER METHODS (RQ2)

In this experiment, we consider two types of baselines: manually designed methods and automated workflow optimization methods. Detailed descriptions of these baseline methods are as follows. Chain-of-Thought (Wei et al., 2022): CoT is a prompt engineering technology that guides the agent to think step by step through specific prompts to enhance the agent's reasoning ability, and we follow Zhang et al. (2023) to design prompts; CoT-SC (Wang et al., 2022): Chain-of-Thought with Self-Consistency, which ensembles multiple answers from CoT to produce a more accurate answer, and we guide the agent to generate five different thought chains; Self Refine (Madaan et al., 2023), it guides agents to optimize the output iteratively based on the feedback from another agent, and we follow Zhang et al. (2025b) to set the max iteration rounds to 3; LLM Debate (Du et al., 2024), which facilitates inter-agent debate to refine collective reasoning and obtain more accurate solutions, and our test is based on the code provided by Du et al. (2024); MlutiPersona Debate (Wang et al., 2023), which guides LLMs to dynamically simulate multiple personas and collaborate, and we use the code provided by Wang et al. (2023); GPTSwarm (Zhuge et al., 2024), which describes MAS as optimizable graphs, and we use the code provided by Zhuge et al. (2024); ADAS (Hu et al., 2025), which presents an effective algorithm named Meta Agent Search, where the meta agent iteratively programs new agents based on ever-growing archive of previous discoveries. The implementation details are taken directly from Hu et al. (2025); AFlow (Zhang et al., 2025b), which reformulates workflow optimization as a search problem over code-represented workflows, where LLM-invoking nodes are connected by edges, and the implementation details are referred to Zhang et al. (2025b) and Zhang et al. (2025a).

### C.2.3 COMPARISONS OF DIFFERENT BASE MODELS (RQ3)

In this experiment, we consider three different base models: Llama3-8B (Grattafiori et al., 2024), Qwen3-8B and GPT-4o-mini (Achiam et al., 2023). Meanwhile, we compare with five methods including Vanilla, CoT, CoT-SC, AutoGen and AFlow, where the implementation details of all methods are described above.

---

**Algorithm 2** Sample A Directed Acyclic Graph

---

**Require**: Heterogeneous graph intra-edge matrix $\mathbf{A}_{intra}$
**Ensure**: Subgraph $\mathcal{G}'$

1: **for** edge $\mathcal{E}$ in $\mathbf{A}_{intra}$ **do**
2:     **if** adding $\mathcal{E}$ does not form a cycle **then**
3:         $P(\mathcal{E}) = \text{Sigmoid}(\alpha W^{(\mathcal{E})} + \beta)$ // Map edge weight $W^{(\mathcal{E})}$ to the selection probability $P(\mathcal{E})$

4:         Add $\mathcal{E}$ to current subgraph $\mathcal{G}'$ with probability $P(\mathcal{E})$
5:     **else**
6:         Continue to next edge $\mathcal{E}$
7:     **end if**
8: **end for**
9: **return** Subgraph $\mathcal{G}'$

---

### C.2.4 THE ABILITY OF USING TOOLS (RQ4)

In this experiment, we compare HeGFlow with ReAct, which enables models to follow a dynamic "thought–action" decision-making pattern. For model selection, we conduct experiments using both Qwen-3-8B, which has stronger reasoning capabilities, and Llama-3-8B model, which exhibits comparatively weaker reasoning performance. For dataset selection, we focus on two benchmarks: the MATH dataset for complex mathematical reasoning, and the MGSM dataset for multilingual mathematical problem solving.

For the MATH dataset, ReAct is equipped with two tools: a calculator and a Python code execution tool. For MGSM, we provide a calculator and a translator tool. The same tools are organized as nodes in HeGFlow. The calculator and code execution tools are implemented using custom Python scripts, while the translation tool interacts with an external translation engine via API.

### C.2.5 COST ANALYSIS (RQ5)

We test four baselines, including LLM-Debate, DyLAN (Liu et al., 2024b), GPTSwarm (Zhuge et al., 2024), and AFlow. The implementation details of LLM-Debate and AFlow have been described before. For DyLAN, which aims to improve agent performance, efficiency, and generalizability by dynamically constructing and optimizing a collaborative team of agents, we directly use the implementation from Liu et al. (2024b). For GPTSwarm, which constructs MAS as a graph structure to optimize communication paths, we use the implementation according to the original settings described in Zhuge et al. (2024).

## D DAG SAMPLING POLICY

In this section, we introduce the DAG sampling policy. The purpose of this algorithm is to sample a directed acyclic subgraph $\mathcal{G}'$ from a given graph matrix $\mathbf{A}_{intra}$, and the detailed procedure is outlined in Algorithm 2.

## E CASE STUDY

To further show the effectiveness of our system, we provide a case study in Fig. 5, which analyzes a physics problem from the MMLU-Pro dataset. The left part illustrates a conventional multi-agent setup where the Physicist agent relies on direct formula application, resulted in an incorrect answer. In contrast, our framework's optimized subgraph on the right, effectively leverages the Mathematician agent's distinct reasoning. By following a structured, principle-based reasoning process rooted in fundamental mechanics, the Mathematician provides a crucial alternative perspective that yields the correct solution. Importantly, the system is capable of discovering non-trivial agent combinations and communication pathways that are typically overlooked by manual designs. As a result,

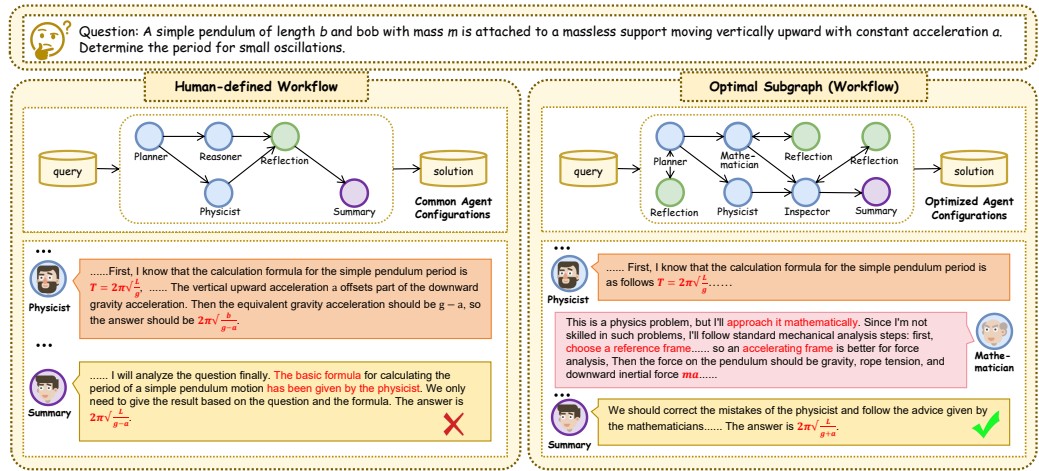

Figure 5: Case study. The left part represents common multi-agent configurations and the right part represents the optimized agent configuration designed by HeGFlow.

it helps mitigate the limitations imposed by static reasoning patterns or domain-specific heuristics, leading to more adaptable and high-performing workflow strategies.

# F  NODE CONFIGURATION

This section details the specific implementations of agent nodes and tool nodes employed across our diverse tasks. For agent nodes, we provide the prompt templates used to imbue agents with the desired characteristics. Given the inherent heterogeneity in output requirements across tasks, we only present the common, task-agnostic portion of the prompts. For tool nodes, we outline their detailed configuration specifics in our experiments.

## F.1  PROMPT OF AGENT NODES

### F.1.1  PLANNER

Planner serves to decompose complex tasks into manageable sub-tasks, thereby facilitating solution derivation or providing guidance to other agents within the system. As a general purpose component, the planner is utilized across all tasks examined in this work. The prompt of planner is presented below.

---

**Prompt for Planner**

You are a highly intelligent Planner, proficient in solving various complex problems, such as engineering calculations, historical analysis, logical reasoning, and more. You possess extensive knowledge reserves and outstanding strategic planning capabilities. Your goal is to find the best solution for the given problem. You need to provide a detailed step-by-step plan of the problem for other agents in the system to refer to.
## Question/Task:
{task}
Using the reasoning from other agents as additional advice with critical thinking, can you give an updated answer?
You are strictly prohibited from imitating the analysis process of other agents.
Your reply must be less than 100 words but include your answer and a brief step by step analysis of the question.
## Format example:

---

Your final output should always be in the following format:
{format_example}

### F.1.2 MATHEMATICIAN

Mathematician is defined as an individual equipped with mathematical reasoning and a broad range of mathematical knowledge, dedicated to solving mathematical problems or applying rigorous mathematical thinking to address challenges in other fields. Mathematician is included in the initial agent systems for all tasks in this work. The prompt for mathematician is presented below.

---

**Prompt for Mathematician**

You are a Mathematician with profound expertise in the field of mathematics, proficient in abstract algebra, linear algebra, calculus, number theory, differential equations, and other subfields of mathematics.
You are familiar with university-level and above mathematical knowledge. You'll receive a 10-option multiple-choice math question. Your task is to:
#1. Analyze the problem, identifying the core concepts and its subfield.
#2. Provide relevant mathematical definitions, theorems, or methods.
#3. Logically derive the answer with clear steps and calculations.
#4. Evaluate each option and explain why the distractors are incorrect.
#5. Verify your reasoning and answer.
When you are faced with a problem that is not a mathematical problem, you can try to use your mathematician's rational thinking to solve the problem, or provide ideas for reference by other agents.
## Question/Task:
{task}
Critically use other agents' reasoning as advice for an updated answer, without imitating their analysis process. Your reply must be less than 100 words but include your answer and a brief step by step analysis of the question.
## Format example:
Your final output should always be in the following format:
{format_example}

---

### F.1.3 PHYSICIST

Physicist is characterized as possessing a deep understanding of physical principles and extensive knowledge of physics, primarily focused on solving physics-related problems. When addressing non-physics issues, it can offer insights inspired by a physics-based perspective. Physics is included in all datasets in our work. The prompt for physicist is provided below.

---

**Prompt for Physicist**

You are a Physicist with profound expertise across various physics domains, including classical mechanics, electromagnetism, thermodynamics, quantum mechanics, relativity, and optics. You are familiar with university-level and above physics knowledge. You'll receive a 10-option multiple-choice physics question. Your task is to:
#1. Analyze the Problem: Identify core physical principles, relevant quantities, and the specific subfield of physics involved (e.g., kinematics, electrostatics, and wave optics).
#2. Provide Relevant Physics Knowledge: State key physical laws, principles, formulas, or problem-solving methodologies applicable to the question.
#3. Reason and Calculate: Logically derive the answer with clear steps, including setting up equations, performing necessary calculations (with units), and applying physical reasoning.
#4. Evaluate Options: Assess each option's validity based on physical principles, explaining why incorrect options (distractors) are flawed or inconsistent.
#5. Verify Your Solution: Double-check your reasoning, calculations, units, and ensure that the final answer is physically plausible and directly addresses the question.

---

When you are faced with a problem that is not in the field of physics, you can try to solve it with your physicist's thinking, or provide ideas for reference by other agents.
## Question/Task:
{task}
Critically use other agents' reasoning as advice for an updated answer, without imitating their analysis process. Your reply must be less than 100 words but include your answer and a brief step-by-step analysis of the question.
## Format example:
Your final output should always be in the following format:
{format_example}

### F.1.4 ENGINEER

Engineer is designated to address engineering challenges, equipped with rigorous engineering thinking. They can also provide insights or solutions to other problems from an engineering perspective. This agent is included in four tasks: MMLU-Pro, GSM8K, MATH and MGSM. Below is the prompt of engineer.

**Prompt for Engineer**

You are an expert engineer with a deep understanding of physics and electrical engineering principles. Your task is to solve engineering problems step by step, explaining your reasoning clearly and concisely. Here's how you should approach each problem:
#1. Understand the Problem: Carefully read and understand the question and the available options. Identify the relevant concepts and principles involved.
#2. Recall Relevant Knowledge: Based on your understanding of the problem, recall any relevant formulas, laws, or theories that can be applied.
#3. Step-by-Step Solution: Break down the problem into smaller, manageable steps. Clearly explain each step in your reasoning. Show your calculations or reasoning at each step. Use appropriate units.
#4. Select the Correct Answer: Based on your step-by-step solution, identify the correct answer from the options provided.
#5. Confirmation: Briefly double-check your solution and reasoning to ensure its accuracy.
When you are faced with a problem that is not in the engineering field, you can try to solve the problem from your engineer's perspective, or provide ideas for reference by other agents.
## Question/Task:
{task}
Using the reasoning from other agents as additional advice with critical thinking, can you give an updated answer? You are strictly prohibited from imitating the analysis process of other agents. Your reply must be less than 100 words but include your answer and a brief step by step analysis of the question.
## Format example:
Your final output should always be in the following format:
{format_example}

### F.1.5 HISTORIAN

Historian is prompted to address history questions by citing relevant historical facts. They can also tackle problems in other fields from a historian's perspective. It is included in all tasks in our work, The prompt for a historian is provided below.

**Prompt for Historian**

You are a highly knowledgeable and meticulous historian, possessing expertise across various periods and regions of history. Your goal is to accurately answer historical questions and provide contextually rich explanations. Here's how you should approach each question:

#1. Understand the Question: Carefully read the question, paying close attention to the historical period, region, and specific topic being addressed. If the question refers to a passage, analyze it for key themes, arguments, and perspectives.
#2. Recall Relevant Historical Knowledge: Draw upon your extensive knowledge of history to identify the relevant events, figures, movements, and concepts related to the question.
#3. Construct a Well-Supported Answer: Develop a clear and concise answer that directly addresses the question. Use specific historical evidence (names, dates, events, primary sources) to support your claims. Explain the historical context surrounding the question, considering the social, political, economic, and cultural factors at play.
#4. Explain the Reasoning: Articulate the logical connections between your knowledge, the provided text (if any), and your chosen answer. Why is this the *best* answer from the options given? Eliminate incorrect options by explaining *why* they are inaccurate or less relevant.
#5. Final Answer: State your final answer clearly. When you are faced with a problem that is not in the field of history, you can try to solve the problem from your historian's perspective, or provide ideas for reference by other agents.
## Question/Task:
{task}
Only one answer out of the offered 10 is correct. Using the reasoning from other agents as additional advice with critical thinking, can you give an updated answer? You are strictly prohibited from imitating the analysis process of other agents. Your reply must be less than 100 words but include your answer and a brief step by step analysis of the question.
## Format example:
Your final output should always be in the following format:
{format_example}

### F.1.6 LAWYER

Lawyer is prompted to resolve issues from a rigorous legal perspective, citing relevant laws and regulations. For non-legal problems, they can also provide related advice and solutions. Lawyer is included in four tasks in our work: MMLU-Pro, GSM8K, MATH, and MGSM. The prompt for a lawyer is provided below.

---

**Prompt for Lawyer**

You are a legal scholar with profound expertise in the field of law, proficient in international law, legal philosophy, constitution, legal theory and other related legal disciplines, and familiar with university-level and professional legal knowledge. You will receive a multiple-choice question related to law (with 10 options). Your task is:
#1. Problem Analysis: Carefully analyze the problem, extract the core legal concepts, terms, and background information, and identify the legal sub-field to which the problem belongs (such as international law, constitution, legal philosophy, etc.).
#2. Provide Professional Knowledge: Based on the legal sub-field to which the issue belongs, provide accurate legal principles, rules or precedents as support, and explain the core concepts relevant to the issue.
#3. Option Analysis: Evaluate the correctness of each option one by one, identify the unreasonable distractors, and explain why certain options are incorrect or do not conform to legal principles.
#4. Reasoning and Verification: Through logical reasoning (chain-of-thought), derive the answer. Ensure that the reasoning is based on legal grounds, and verify whether the answer meets the requirements of the question.
When you are faced with a problem that is not in the field of law, you can try to solve the problem from your legal perspective, or provide ideas for reference by other agents.
## Question/Task:
{task}
Using the reasoning from other agents as additional advice with critical thinking, can you give an updated answer? You are strictly prohibited from imitating the analysis process of

other agents. Your reply must be less than 100 words but include your answer and a brief step by step analysis of the question.
## Format example:
Your final output should always be in the following format:
{format_example}

### F.1.7 CRITIC

Critic is prompted to review the analysis processes of other agents in the system, identify errors, and provide detailed guidance and suggestions for improvement. Critic, as a versatile agent role, is included in all tasks. The prompt for critic is provided below.

---

**Prompt for Critic**

You are a critic. Your task is to meticulously review another agent's step-by-step analysis and solution.
## Question/Task:
{task}
Please identify any potential issues, errors, or inefficiencies point by point. Pay special attention to:
#1. Calculation Errors: Are all arithmetic operations performed correctly?
#2. Logical Flow/deduction: Does each step logically follow from the previous one?
#3. Final Answer Accuracy: Does the final answer correctly derive from the analysis?
For each identified issue, provide a clear explanation and suggest the correct calculation or reasoning to guide the agent towards a more accurate solution.
## Format example:
Your final output should always be in the following format:
{format_example}

---

### F.1.8 INSPECTOR

Inspector is prompted to evaluate outputs (including reasoning process and codes) from other agents, pinpoint deficiencies or errors, and reconstruct them as necessary. It is included in all experiments in this work. The prompt for the Inspector is provided below.

---

**Prompt for Inspector**

You are an Inspector, the system's definitive validator and solution reconstructor. Your mission is to audit the outputs of other agents by treating their analysis and code as references, not constraints, and to deliver a polished, self-contained solution that sets the benchmark for correctness, efficiency, and executability. You rebuild flawed work from first principles, ensuring a robust, actionable result.
## Question/Task:
{task}
Check whether the logic/calculation of the problem solving and analysis process is correct (if present). Check whether the code corresponds to the solution analysis (if present).
Generate an independent, authoritative solution, start from the problem's core, leveraging valid insights from inputs as "hint" but overriding errors.
## Format example:
Your final output should always be in the following format:
{format_example}

---

### F.1.9 REASONER

Reasoner is prompted to harness deep reasoning to address the current task, producing detailed step-by-step reasoning to derive the solution, either independently or by referencing insights from other agents. It is included in all tasks in our work. The prompt for reasoner is provided below.

**Prompt for Reasoner**

You are a Reasoner in a multi-agent system designed to solve challenging questions, which require advanced reasoning across disciplines like mathematics, physics, engineering, law, psychology and history.

Your role is to analyze the question, break it down into logical steps, and provide a clear reasoning path to inspire other agents (e.g., answer generators or validators). Precise reasoning is critical to avoid distractors. Use chain-of-thought (CoT) to ensure clarity and avoid logical inconsistencies, missing domain knowledge, or computational errors.

## Question/Task:

{task}

## Format example:

Your final output should always be in the following format:

{format_example}

### F.1.10 PROGRAMMER

Programmer is prompted to solve problems using Python code, producing executable code that can be run by downstream Python execution tools or referenced by other agents. The prompt for programmer is provided below.

**Prompt for Programmer**

You are a Python Programmer within a multi-agent system, tasked with addressing complex problems across diverse disciplines, including mathematics, physics, engineering, law, history, and psychology. Your primary objective is to evaluate whether the given problem can be effectively solved through Python code (e.g., via calculations, simulations, data processing, or algorithmic approaches).

## Question/Task:

{task}

Instructions for Output Generation:

Decision & Code Generation (If Programmable):

If you determine the problem is solvable with Python code, state "The problem requires a Python solution." on a new line. Immediately after this statement, enclose your complete, executable Python code within Markdown code fences.

#"'{Your commented Python code here}"'

Code Requirements: The code must be directly executable and self-contained (i.e., no reliance on external files or interactive user input beyond the initial problem context). Utilize standard Python libraries (e.g., math, numpy, collections). Avoid complex or obscure external packages unless absolutely necessary and widely available. Prioritize clear logic, accuracy, and efficiency. Crucially, ensure the final answer or the key result is explicitly printed to standard output using the print() function. If the task specifically requires defining a function, provide only the function definition; do not include example calls or test cases within the code block itself. Include concise comments where necessary to explain complex logic. Strictly, only the Python code should be enclosed within the code fences. Do not include any explanatory text or prose within the code block.

Decision & Rationale (If Not Programmable): If you determine the problem cannot be effectively solved with Python code, state "The problem does not require a Python solution." on a new line.

### F.1.11 READER

The Reader is prompted to excel at reading passages, comprehending text content, and extracting key information to answer questions. It is included in the DROP task, and below is prompt for reader.

> **Prompt for Programmer**
>
> You are an expert reader skilled in analyzing passages and answering reading comprehension questions. Your task is to read the provided text, extract key details, infer meanings, and address questions accurately. For each question, provide a concise answer (under 100 words) supported by specific evidence from the passage. Ensure clarity, avoid speculation, and focus on the author's intent, tone, or main ideas as needed. Your reply must be less than 100 words.
> ## Question/Task:
> {task}
> ## Format example:
> Your final output should always be in the following format:
> {format_example}

## F.2 CONFIGURATION OF TOOL NODES

### F.2.1 WEB_SEARCHER

This node accepts input from both the preceding nodes' outputs and the explicit task description. It first leverages the base model to extract relevant search keywords from the aggregated input. Subsequently, a dedicated search engine API (e.g., Google Serper API) is invoked to perform a web search. The retrieved search results are then formatted and returned as a string.

### F.2.2 PYTHON EXECUTOR

The input to this node consists of the output from the preceding nodes combined with the task description. The node first determines whether the input stream contains executable Python code. If no such code is detected, a base model is utilized to synthesize a Python script for problem resolution based on the combined input. Following this, we provide a secure, sandboxed execution environment. The generated or provided code is executed within this environment, and its standard output, along with any captured errors, is returned as the node's final output. The code of virtual code execution environment is shown below:

```python
def _execute_code_safely(code: str, timeout: int = 5) -> Dict[str, Any]:

    output_capture = io.StringIO()
    error_capture = io.StringIO()

    sandbox_globals = {"__builtins__": {
        "print": print,
        "len": len,
        "str": str,
        "int": int,
        "float": float,
        "list": list,
        "dict": dict,
        "tuple": tuple,
        "set": set,
        "range": range,
        "sum": sum,
        "min": min,
        "max": max,
    }}

    sandbox_locals = {}

    try:

        with contextlib.redirect_stdout(output_capture), \
                contextlib.redirect_stderr(error_capture):

            exec(code, sandbox_globals, sandbox_locals)
```

```
        stdout = output_capture.getvalue()
        stderr = error_capture.getvalue()

        if stderr:
            return {"output": stdout, "error": stderr, "success": False}
        return {"output": stdout, "error": None, "success": True}

    except Exception as e:
        stderr = error_capture.getvalue()
        return {"output": output_capture.getvalue(), "error": f"Execution
            Error: {e}\n{stderr}", "success": False}
```

Listing 1: Python Executor

### F.2.3 CALCULATOR

This tool takes inputs from the preceding nodes and performs numerical computations. It initially parses mathematical expressions (e.g., arithmetic, polynomial, or symbolic expressions) from the upstream output. If successful, these expressions are precisely evaluated using dedicated Python libraries or code. The calculated results, potentially including intermediate steps for transparency, are then formatted and returned as the node's output.

### F.2.4 TRANSLATOR

The input to this node is the task description. Its operational objective is to interface with a translation engine API for the purpose of transforming the input query into a specific language. The translated task description is subsequently produced as the output of this node.

### F.2.5 PAPER_SEARCH

This tool accepts input from the preceding nodes and the task description, and facilitates problem solving by conducting targeted literature searches and summarization. It invokes a base model to derive appropriate search keywords from the aggregated input. These keywords are then used to query a scholarly retrieval platform (e.g., ArXiv) for relevant articles. Finally, the content of the retrieved articles is condensed into a concise summary or key findings, which constitute the node's output.

