# OpenReview forum: "A Heterogeneous Graph Optimization Perspective for Multi-Agent System Workflows"
_ICLR.cc/2026/Conference — ICLR 2026 Conference Withdrawn Submission_

### Official Review · Reviewer_UsaC · 2025-10-30

**Soundness:** 2
**Presentation:** 1
**Contribution:** 3
**Rating:** 4
**Confidence:** 2

**Summary:**

This paper introduces HeGFlow, transforming the MAS optimization problem from a discrete search problem relying on meta-agent generation into a continuous adjacency matrix optimization problem. HeGFlow models MAS as heterogeneous graphs, with agent nodes and tool nodes, optimizing from coarse to fine through a two-phase approach. Through five research questions, experiments demonstrate that HeGFlow: (1) achieves robust performance across diverse domains, (2) outperforms existing manual and automated methods, (3) generalizes across different model architectures and scales, (4) exhibits superior tool utilization capabilities, and (5) maintains significantly lower token consumption while delivering better results.

**Strengths:**

- **Novel Formulation**: The heterogeneous graph representation of MAS workflows is intuitive and elegant. Modeling agents and tools as distinct node types with learnable edge weights provides a principled framework for workflow optimization.
- **Comprehensive Benchmarks**: The paper evaluates HeGFlow across diverse benchmarks (MMLU-Pro, DROP, GSM8K, MATH, MGSM) covering multiple domains, demonstrating robust performance gains.
- **Cost Efficiency**: The two-stage optimization strategy and sampling policy address the computational challenges of exploring large search spaces. The cost analysis shows HeGFlow achieves better performance with lower token consumption compared to baselines.

**Weaknesses:**

- **Matrix Formulation**: The notation $A=A_{\text{intra}}\cup A_{\text{inter}}$ (Section 2.1) is mathematically imprecise and creates cascading confusion throughout the paper. The union symbol is inappropriate for matrices with different semantic meanings (A_intra controls intra-round connections including self-loops; A_inter controls inter-round connections from round $k-1$ to $k$). More critically, the paper never clarifies how these matrices are actually used. This ambiguity makes the entire method uninterpretable.
- **Parameter K in Training and Execution**: The paper states "our system performs K iterative rounds per query" and Figure 1 shows "K rounds execution," clearly indicating that K-round execution happens during inference, not just training. This may bring about unfairness in testing? But I'm not sure if there is a difference in specific training and execution here, which is a place that needs clearer expression.
- **Critical Experiments and Analyses Missing**: The paper lacks essential experiments: (1) Ablations about K and parameters in sampling policy. (2) Insufficient tool usage analysis. Section 3.4 reports only accuracy gains without explaining how tools are used, missing statistics.
- **Presentation Quality Issues**: Algorithm 1 line 14 missing closing parenthesis. Figures 3-4 are raster images rather than vector graphics (like pdf or svg). Figure 2 uses unexplained notation "W R E D" without definition, and lacks color legend for matrix visualization.

**Questions:**

- Does the system execute K iterative rounds during both training and inference? Please clarify: (a) the K value(s) used in your experiments, (b) whether training and inference use the same K, (c) the exact inference procedure showing how $A_intra$ and $A_inter$ orchestrate K-round execution.
- The current approach appears to optimize a single workflow per task/domain. Could the framework be extended to dynamically select or adapt workflows based on query characteristics?
- While comparisons with GPTSwarm, ADAS, and AFlow are reasonable, more recent work like ScoreFlow and others on agent workflow optimization have emerged. Can you comment on how HeGFlow would compare to these newer approaches?
- HeGFlow appears to use a fixed, predefined set of nodes. Are the node types and quantities manually predefined for each task, or is there a principled way to determine the initial node set? How sensitive is the framework to the initial node selection? Can the framework dynamically add new agent types or tools during optimization, or is the node set fixed throughout training?

---

### Official Review · Reviewer_WUTZ · 2025-11-01

**Soundness:** 3
**Presentation:** 3
**Contribution:** 3
**Rating:** 4
**Confidence:** 4

**Summary:**

The paper introduces HeGFlow, a novel framework for Multi-Agent System (MAS) workflow optimization. It models MAS as a heterogeneous graph, where agents, tools, and interactions are represented as nodes and edges. The problem of automating MAS workflows is reformulated as adjacency matrix optimization, which is then efficiently explored using a two-stage training strategy with a subgraph sampling policy. The approach aims to improve the efficiency of MAS workflow generation and has been validated across multiple benchmarks, demonstrating that it outperforms both manually designed and automated workflow optimization methods, particularly when using smaller models that achieve comparable results to larger ones.

**Strengths:**

S1 (Novel approach): The introduction of a heterogeneous graph framework to model MAS workflows is creative. It provides a fresh perspective by optimizing the graph structure rather than focusing on just agent behavior.

S2 (Scalability): The two-stage training approach—coarse-grained node selection followed by fine-grained edge optimization—is an effective strategy to handle the vast search space, ensuring both exploration and exploitation.

S3 (Empirical results): The experiments across multiple domains demonstrate that HeGFlow consistently outperforms existing methods, especially in scenarios that require agent collaboration and reasoning.

**Weaknesses:**

W1 (Limited to MAS): The proposed approach is specialized for MAS workflows and does not yet demonstrate its applicability to other types of multi-agent systems or domains outside the evaluated ones (e.g., complex real-time systems).

W2 (Lack of flexibility in model choice): While HeGFlow performs well with smaller models, it remains unclear how well it adapts to more diverse or non-standard models, particularly deep models where traditional LLMs may not be directly applicable.

W3 (Over-complication of graph structure): The heterogeneous graph structure, while theoretically sound, might add unnecessary complexity in certain simple tasks. Simplified alternatives could be more efficient for tasks with minimal inter-agent interaction.

W4 (Interpretability of results): Although the framework improves performance, it does not provide sufficient insights into how the learned graph structure affects agent behavior in specific tasks. More detailed analysis could improve understanding of what drives success in different scenarios.

W5 (Implementation details): Some of the experimental results are difficult to interpret due to lack of clarity in how the metrics are calculated, particularly in the context of cost analysis and token consumption (e.g., in Figure 4).

**Questions:**

Q1. Can HeGFlow be generalized to workflows that require real-time agent decisions or highly dynamic environments? What would be needed to adapt it to such scenarios?

Q2. The two-stage graph optimization process seems resource-intensive. Can you provide more detailed performance benchmarks in terms of training and inference times, especially on larger datasets?

Q3. How would the approach perform with larger, more complex models (e.g., GPT-4 or similar)? Could the framework still operate efficiently, or would the benefits diminish?

Q4. While tool integration is a strong point, how does HeGFlow handle the quality and consistency of external tools when they fail or produce unreliable outputs?

Q5. In Figure 5, you show improvements in MAS configuration. Could you expand on how the graph-based approach identifies “non-trivial agent collaborations” and whether this method can be interpreted or explained?

---

### Official Review · Reviewer_B8dt · 2025-11-01

**Soundness:** 2
**Presentation:** 3
**Contribution:** 2
**Rating:** 4
**Confidence:** 4

**Summary:**

This paper proposes HeGFlow, which models MAS workflow generation as a heterogeneous graph adjacency matrix optimization problem. Nodes include agents (Planner, Mathematician, Physicist, etc.) and tools (Calculator, Python Executor, etc.), with edges categorized as intra-round (Aintra) and inter-round (Ainter), while self-loops represent reflection. A two-stage training approach is introduced: Stage-1 performs coarse-grained node selection via Lsparse and nuclear norm regularization, while Stage-2 refines edge weights on the retained nodes. The sampling strategy adopts UCB to balance exploration-exploitation. Experiments on MMLU-Pro (six subdomains), DROP, GSM8K, MATH, and MGSM show: HeGFlow (Qwen3-8B) achieves an average of 69.3, outperforming CAMEL (62.2) and AutoGen (56.0); on GSM8K, Llama3-8B improves by 20.8%; in tool-use scenarios (MGSM/MATH), it surpasses ReAct.

**Strengths:**

1. **Structured problem formulation**: Abstracting MAS workflows as graph optimization with explicit distinctions between intra-round/inter-round interactions and reflective self-loops provides a trainable framework with clearer optimization objectives compared to natural language-based historical experience.

2. **Well-motivated two-stage training strategy**: The staged optimization of nodes and edges theoretically reduces the search space and accelerates convergence, potentially more efficient than full end-to-end matrix training, although direct comparative experiments are missing.

3. **Theoretically grounded sampling strategy**: The UCB-inspired scoring mechanism (Equation 3) explicitly incorporates exploration terms and diversity rewards, theoretically helping to escape local optima, though empirical validation through ablation studies is needed.

4. **Consistent cross-model performance**: Experiments cover models of different scales (Llama3-8B/Qwen3-8B/GPT-4o-mini) and multi-domain datasets, showing performance improvements over AutoGen/CAMEL with the same base model.

5. **Quantitative evaluation of tool orchestration**: In MGSM's multilingual scenarios and MATH's computational scenarios, HeGFlow achieves gains over ReAct by automatically learning tool invocation sequences, although attribution remains unclear.

**Weaknesses:**

1. **Fundamental flaws in performance claims and experimental design**

- **Table 1 comparison confounds two variables, failing to support the "small model matches large model" claim**:
  - Experimental design issue: Table 1 compares "HeGFlow (Qwen3-8B as base model, 8B parameters) vs GPT-4o/GPT-4-Turbo (closed-source large models)", simultaneously changing two factors: (1) workflow design (HeGFlow vs 5-shot CoT); (2) model type (open-source small model vs closed-source large model). It's impossible to determine whether the improvements come from workflow optimization or from Qwen3-8B's inherent capabilities being comparable to or superior to GPT-4o in these domains.
  - Qwen3-8B's base capabilities on benchmarks are severely underestimated: Table 1 reveals a critical fact—Qwen3-8B (5-shot CoT) already achieves 75.1 on the Math subdomain, only 1 point behind GPT-4o (5-shot CoT) at 76.1. This indicates that despite having fewer parameters, Qwen3-8B's capabilities on specific benchmark tasks are already close to or matching closed-source large models. Although the paper uses 5-shot CoT for all baselines (Appendix C.2.1), the similar performance between Qwen3-8B and GPT-4o on Math shows that on this specific benchmark, their base capabilities are comparable. In this case, the test-time gain from HeGFlow naturally exceeds inference from similarly capable models—but this cannot support the generalized conclusion that "smaller models surpass larger models."

2. **Lack of analysis and interpretability of learned workflows**

- **The paper fails to present learned optimal workflow structures—unacceptable for a workflow optimization paper**:
  - Only 1 case study is provided (Appendix E, Figure 5): a workflow visualization for an MMLU-Pro physics problem.
  - Missing content: (1) Comparison of learned graph structures across domains (Math vs History vs Law); (2) Workflow differences across task types (reasoning vs reading comprehension vs multilingual); (3) Which nodes are retained/pruned after Stage-1; (4) Edge weight distribution heatmaps for different domains; (5) Evolution of graph structure during training.
  - Why this is a fundamental flaw: Unlike black-box models such as neural networks, workflows are not systems lacking interpretability—they are fully interpretable, analyzable, and verifiable flow diagrams. In reasoning tasks (without complex environment interactions), the base model's capabilities almost completely determine the range of performance differences between frameworks. Any performance improvement must come from workflow design differences (agent invocation order, information flow, tool usage strategies, etc.), all of which are observable, reproducible, and verifiable. Therefore, claiming performance gains from workflow optimization without showing the learned workflow structure is equivalent to claiming a new algorithm without disclosing algorithm details—unacceptable in a methods paper. The paper must provide complete workflows for at least 5-10 representative tasks, enabling readers to: (1) understand why HeGFlow succeeds on Math; (2) diagnose why it fails on History; (3) verify that improvements indeed come from workflow rather than other factors (e.g., more operators, stronger base model).

- **Complete absence of Ablation Studies**:
  - Particularly critical is the lack of controlled experiments with aligned operator sets: HeGFlow uses 11 agent nodes + 5 tool nodes (Appendix F), apparently more than baseline methods. When HeGFlow outperforms baselines, it's impossible to determine whether improvements come from: (1) the workflow optimization algorithm, or (2) richer operator variety. Ablation experiments with HeGFlow using aligned operator libraries (matching AFlow's operator set) are essential to isolate the impact of operator count.

3. **Potential flaws in method design**

- **Node uniqueness constraint limits same-round operator reuse**:
  - The adjacency matrix modeling assumes each agent/tool is a unique node (V = Vagent ∪ Vtool, Line 250). Combined with the DAG constraint (Algorithm 2), the same agent cannot be invoked multiple times within a round. For example: Self-Consistency requires 5 parallel CoT paths, or divide-and-conquer tasks require multiple Mathematician agents processing sub-problems in parallel, but HeGFlow cannot directly express such patterns.

- **Circular dependency issue in Lsparse definition (Equation 5, Lines 706-712)**:
  - wi is defined as "for inefficient nodes, assign larger wi; for efficient nodes, assign smaller wi," but how is "efficiency" defined? Based on historical correctness rate, sampling frequency, or average edge weight µvi? If based on µvi, this creates a circular dependency (edge weight optimization depends on wi, while wi depends on edge weights), potentially leading to training instability or trivial solutions (all wi → 0). The paper provides neither wi computation details nor ablation experiments.

**Questions:**

1. **Regarding Table 1 comparison**: The current comparison simultaneously changes both workflow design and base model. How can you prove that performance gains come from workflow optimization rather than LLM's base capabilities on these benchmarks?

2. **Regarding operator library fairness**: HeGFlow uses 11 agent types + 5 tool types (Appendix F). How much of the performance gain comes from the workflow optimization algorithm versus richer operator variety? A brief discussion would be helpful.

3. **Where are the learned workflows?**: The paper only shows one case (Appendix E, Figure 5). Can you provide complete workflow structure comparisons across different domains?

4. **Impact of node uniqueness constraint**: The adjacency matrix modeling assumes each agent/tool is a unique node. How does this handle scenarios requiring repeated invocation of the same operator? Does this limit the method's expressive power?

5. **How is wi in Lsparse defined** (Equation 5): The paper states "assign larger wi to inefficient nodes, smaller wi to efficient nodes," but what is the specific definition of "efficiency"? If wi depends on edge weight µvi, how do you avoid circular dependency (edge weight optimization depends on wi, while wi depends on edge weights) leading to training instability?

---

### Note · Authors · 2025-11-23

I have read and agree with the venue's withdrawal policy on behalf of myself and my co-authors.